# Patients’ Clinical and Psychological Status in Different COVID-19 Waves in Italy: A Quanti-Qualitative Study

**DOI:** 10.3390/healthcare11182477

**Published:** 2023-09-06

**Authors:** Martina Vigorè, Andrea Steccanella, Marina Maffoni, Valeria Torlaschi, Alessandra Gorini, Maria Teresa La Rovere, Roberto Maestri, Maurizio Bussotti, Sergio Masnaghetti, Francesco Fanfulla, Antonia Pierobon

**Affiliations:** 1Psychology Unit, Istituti Clinici Scientifici Maugeri IRCCS, 27040 Montescano, Italy; 2Istituti Clinici Scientifici Maugeri IRCCS, Milano-Camaldoli, 20138 Milan, Italy; 3Dipartimento di Scienze Cliniche e di Comunità, Università degli Studi di Milano, 20122 Milan, Italy; 4Department of Cardiology, Istituti Clinici Scientifici Maugeri IRCCS, 27040 Montescano, Italy; 5Department of Biomedical Engineering, Istituti Clinici Scientifici Maugeri IRCCS, 27040 Montescano, Italy; 6Cardiorespiratory Rehabilitation Unit of Milano Institute, Istituti Clinici Scientifici Maugeri IRCCS, 20138 Milan, Italy; 7Department of Medicine and Cardiopulmonary Rehabilitation, Istituti Clinici Scientifici Maugeri IRCCS, 21049 Tradate, Italy; 8Respiratory Function and Sleep Medicine Unit, Istituti Clinici Scientifici Maugeri IRCCS, 27010 Pavia, Italy

**Keywords:** COVID-19, pandemic waves, functional, distress, PTSD, depression

## Abstract

Background: COVID-19 waves have been characterized by different clinical manifestations, a decrease of functional abilities, and the presence of psychological symptoms. The aims of this study were to investigate differences in physical and psychological symptoms during the three Italian waves of the coronavirus pandemic. Methods: Patients undergoing a functional, cardiological and pneumological check-up follow-up at ICS Maugeri Institutes, 2–3 months after recovery from COVID-19 were consecutively recruited to participate in the study, completing a quanti-qualitative questionnaire about anxiety, depression, PTSD symptoms, and personal resources. Results: 104 patients were recruited: 44 and 60 during the first and second/third pandemic waves, respectively. Physical comorbidities were more present in the second/third waves compared to the first one, while no significant differences were found in anxious and depressive symptoms, which were significantly higher than normal during the three waves; PTSD symptoms were reported by 56.3% of patients. Family, social support, and a positive mindset were described as resources to cope with the COVID-19 burden. Negative affects arose during outbreaks (panic) and the disease (fear), while positive affect (joy) characterized only the follow-up period. Conclusion: This study shows how psychophysical symptoms might change during the pandemic waves and highlights the importance of protective factors to balance the subjective distress.

## 1. Introduction

As known, coronavirus disease (COVID-19) is an infectious disease caused by the SARS-CoV-2 virus. Everybody can be infected at any age and some of them develop a severe disease requiring relevant medical assistance during their illness and recovery. Older people and patients with comorbidities (e.g., cardiovascular disease, diabetes, respiratory disease, cancer) have an increased risk of developing serious illnesses due to COVID-19 disease [1].

In Italy three pandemic waves were identified: the first wave (1 March 2020–15 April 2020), the second wave (15 October 2020–15 December 2020) and third one (1 March 2021–15 April 2021). The first wave was characterized by an overwhelmed and unprepared health care system, with a higher ICU admission and mortality percentage and by strict restrictions in all the Nation, conversely, in the second and in the third waves there were a progressive reduction of mortality, ICU admission and better organizational patients’ tracking and management [2].

Regardless of differences and severity of symptoms of the three pandemic waves, all of them have left significant post-acute sequelae in many patients [3], including, but not limited to, dyspnea and fatigue, ageusia, cough, anosmia, headache, confusion, and joint pain [3,4]. Post-infection physical and emotional consequences derived from these and other pathological conditions have significantly affected the patients’ daily life activities, consequently decreasing quality of life [5,6,7].

Considering genders, there are discordant data about the incidence of COVID-19. Anyway, the differences found (e.g., men seem more likely to develop the condition, have a worse prognosis, twice as likely to die) could be related to biological attributes, including hormonal, immune, and inflammatory response to infection but also to social and behavioral characteristics. Studies have shown the presence of greater concern by women with respect to the severity of the pandemic and, in turn, they are more adherent to public containment measures and restrictions. In addition, men have a higher prevalence of high-risk behaviors and hold jobs that expose them more to the risk of infection too [8].

Focusing on a psychological and neuropsychiatric perspective, post-infected individuals seem to have an increased risk of developing anxious and depressive symptoms [9,10,11], as well as decreased sleep quality [12,13], and increased generalized distress mainly related to the social limitations imposed by the pandemic [9].

A significant association between infection and Post-Traumatic Stress Disorder has been also observed [10,11,12], especially in women, in low-educated patients, and in individuals with high levels of anxiety and low perceived emotional support [14]. Shanbehzadeh et al. (2021) [7] also showed that one-third of COVID-19 patients who did not report any psychological symptom during the infection period, reported a form of psychological sufferance 6 months after it. Finally, regarding the correlation between physical and psychological symptoms in post-COVID patients, it has been observed that fatigue [15], but also anxiety and depression [16] correlate with PTSD. Anxiety and depression also correlate with dyspnea and asthenia [17], while reduced mobility significantly affects quality of life [18]. In addition, the severity of physical symptoms affects the level of psychological distress and quality of life [7,15,17,18].

To our knowledge, few studies have investigated the differences in physical symptoms in the different pandemic waves and the changes in the patients’ emotional experiences. Specifically, it is possible that difficulties, resources and emotions experienced by patients could be different according to differences in the COVID-19 disease experience itself; however, little it is known other than suggestions from everyday experience. Thus, the main aim of this study is to investigate the differences in psychophysical symptoms between waves, and the second one is to analyze the different emotions perceived by patients in different moments of the COVID-19 spread.

## 2. Materials and Methods

### 2.1. Participants and Procedure

In this study were considered eligible all subjects (age ≥18) consecutively undergoing cardiological and pneumological check-up follow-up at ICS Maugeri—Montescano (Pavia), Tradate (Varese), Pavia and Milano institutes, from 2–3 months to one year after recovery from COVID-19.

All participants suffered from COVID-19 in Italy during the first wave (1 March 2020–15 April 2020) or second and third ones (15 October 2020–15 December 2020; 1 March 2021–15 April 2021, respectively) [2], not necessarily requiring hospitalization.

The exclusion criteria were acute severe clinical conditions (e.g., severe chronic heart failure, respiratory failure, etc.), no Italian education or relapse into illiteracy, severe visuo-perceptive deficits, lack of motivation or refusal to underwent to the evaluation.

The sample was composed by 104 patients, who were enrolled on a voluntary basis, after being properly briefed and signing the informed consent form. No kind of remuneration was provided.

The study was approved by the Institutional Review Board and Central Ethics Committee of the ICS Maugeri SpA SB (CEC) (approval number: CEC N. 2450, 21/07/2020).

Patients’ COVID-19 disease data, clinical management and comorbidities (T_0_), extrapolated from the computerized medical record, were collected for this study. Upon the clinical check-ups (T_1_) patients underwent clinical, functional and psychological evaluation. Data were analyzed considering the whole sample and according to the first and second/third pandemic waves.

Functional evaluation included electrocardiogram (ECG), 2D-Echocardiography, 6-min walking test (6MWT), arterial blood gas determination.

For the psychological assessment, at T1, patients were requested to fulfill an ad hoc questionnaire (see Appendix A) comprising the following tests: National Stressful Events Survey PTSD Short Scale (NSESSS), Patient Health Questionnaire-9 (PHQ-9) and Generalized Anxiety Disorder-7 (GAD-7). Moreover, participants were requested to state difficulties and resources to cope with pandemic and to refer emotions experienced in the different phases of pandemic.

The quanti-qualitative psychological evaluation was done until one year from the acute phase referring to the first or the second/third wave of the COVID disease.

Thus, in brief, we retrospectively retrieved clinical data of the COVID disease (T_0_) from the computerized clinical records; at follow-up we collected both clinical-functional data and qualitative and quantitative psychological ones.

### 2.2. Materials

Patients participating in this study were firstly asked to fill out the Socio-anagraphic Schedule investigating socio-demographic variables, some clinical data and risk factors. Clinical data regarding risk factors, comorbidities and clinical management of COVID disease such as the necessity of hospitalization and the type of treatment received [Oxygen Therapy (OT), Continuous Positive Airway Pressure (CPAP), Non Invasive Ventilation (NIV) or invasive mechanical ventilation (IMV)] were retrospectively retrieved from computerized clinical records.

#### 2.2.1. Functional Evaluation

The 6MWT is a self-limited test used to measure functional exercise abilities. In this evaluation the person is asked to walk as fast as possible compatible with his clinical condition for a time of 6 min, measuring the meters traveled [19].

#### 2.2.2. Psychological Evaluation—Quantitative Part

National Stressful Events Survey PTSD Short Scale (NSESSS) is a validated self-report 9-item questionnaire that assesses the severity of Post-Traumatic Stress Disorder in patients older than 18 years after an extremely stressful event or experience (following DSM criteria). The questionnaire consists of 9 items and it organizes the severity of the responses on a 4-point Likert scale. The final score has a range between 0–36. A higher score indicates greater severity of Post-Traumatic Stress Disorder. The clinician has to calculate and use the average total score. The mean total score converts the overall score to a 5-point scale, which allows the clinician to rate the severity of Post-Traumatic Stress Disorder in subject as none (0), mild (1), moderate (2), severe (3), or very severe (4) [20,21].

Patient Health Questionnaire-9 (PHQ-9) is a scale used to determine the diagnosis, severity, and subsequent monitoring of depressive illnesses of the patient. It is divided into nine sub-items, to identify depressive symptoms within the last two weeks (following DSM criteria). The final score has a range between 0–27. It is divided into clinical variability ranges according to a continuum of symptom severity, where scores of 5, 10, 14, and 19 are considered cut-offs for subthreshold, mild major, moderate major, and severe major depression, respectively [22,23].

Generalized Anxiety Disorder-7 (GAD) is a questionnaire built to measure the severity of anxiety symptoms in the previous two weeks. The questionnaire consists of 7 items and it organizes the severity of the responses on a 4-point Likert scale. The scores range from 0 to 21, where scores of 5, 10, and 15 are considered cut-offs for mild, moderate, and severe anxiety, respectively [23,24].

#### 2.2.3. Psychological Evaluation—Qualitative Part

Besides the overmentioned questions, participants were asked to state resources to cope with the pandemic and possible work difficulties experienced after the recovery. Moreover, they were requested to provide one to three emotions experienced in the following phase of the pandemic: (1) outbreak in China, (2) outbreak in Italy, (3) during their own disease, (4) at the present time (follow-up). See Appendix A.

### 2.3. Statistical Analysis

In this study, the sample was divided into two groups based on the patient’s infection surge of coronavirus. We compared the first wave with the second and third waves, as the clinical conditions and characteristics of patients who contracted the disease after the first wave can be considered similar and generally less severe than those who faced the disease from COVID-19 at the beginning of pandemic [2]. Thus, all analyses compare the first wave versus the second/third one.

Descriptive statistics are reported as mean ± SD for continuous variables and as numbers (N) and percentage for discrete variables. Between-group comparisons (first wave vs. second/third wave) were carried out by the Mann-Whitney U test and the Chi-squared test for continuous and categorical variables, respectively.

To investigate whether an association between psychological evaluation scores and clinical variables was present, correlation analysis was carried out (Spearman’s correlation coefficient r).

The association between couples of variables was assessed by the Spearman’s correlation coefficient.

For GAD, PHQ, and NSESS scores, the frequency distributions of dichotomized items (<10 vs. ≥10) in the study population were compared with the respective frequency distribution of the normative population using the Chi-square test or the Fisher exact test, as appropriate. The null hypothesis was that the relative frequency of each category equals the normative frequency for that variable.

All statistical tests were two-tailed and a *p*-value < 0.05 was considered statistically significant. When appropriate, false discovery rate was controlled at 5% using the Benjamini-Hochberg method. All analyzes were performed using the SAS/STAT statistical package, version 9.4 (SAS Institute Inc., Cary, NC, USA).

#### Text Analysis

Concerning the open-ended questions, we conducted a descriptive text analysis, making comparison between first and second/third waves.

Firstly, the authors analysed the terms or short sentences identified by patients as resources to cope with the pandemic or as difficulties experienced in returning to work.

Themes categorization was independently conducted by two reviewer (MV, AP) and inconsistences were managed by a third reviewer with specific expertise in qualitative analysis (MM). Only themes reaching full agreement was considered as relevant results. Finally, emerged themes has been organized into categories and frequencies which are displayed through bar charts.

Secondly, emotions connected to the different stages of pandemic were also collected and categorised and quantified. Emotions were preliminary read to purge typing errors and semantically irrelevant words. Specifically, adjectives and verbs were turned into nouns of emotion, and plural forms were turned into singular ones. Then, the authors (MM and AP) determined conceptual categories according to a theory-driven approach, that is Plutchik’s Wheel of Emotions Theory [25,26] in order to unveil possible differences in the emotional experiences of COVID-19 patients over time. Briefly, according to this author, emotions are the result of an evolutionary process in which events, cognitions, feelings, and actions are interconnected. It is possible to distinguish eight primary emotions displayed in a circle on the bases of semantic studies unveiling semantic similarities and differences between emotional terms used by individuals. Thus, in the classical flower-shaped representation, the semantic proximity is represented as spatial proximity, so that the petal of joy is located beside the emotion of trust and in the opposite position with respect to sadness. The basic emotions are the following: (a) Joy, ranging from serenity to ecstasy; (b) Sadness ranging from pensiveness to grief; (c) Trust ranging from acceptance to admiration; (d) Disgust ranging from boredom to loathing; (e) Anger ranging from annoyance to rage; (f) Fear ranging from apprension to terror; (g) Surprise ranging from distraction to amazement, (h) Anticipation ranging from interest to vigilance. Following this representation and theory, it is possible to unveil secondary emotions and different levels of intensity, so that this theoretical framework may be considered promising in studies considering clinical settings [27].

All emotion categories were discussed until a consensus was reached and all authors provided final feedback on the identified categories. In case of doubts, the Italian Collins Thesaurus was consulted for looking for synonyms. Differences between the frequencies of terms between different waves were calculated by Fisher exact test, two tails (*p* < 0.05) and the flower-shaped representation of emotions were arranged through the Python module “PyPlutchik”, freely available on the Github repository [27].

## 3. Results

### 3.1. Quantitative Results

Overall, 112 patients were enrolled: of these 8 were excluded because the psychological assessment was conducted after 1 year from infection (Figure 1). Then, 104 patients were recruited: of these, 44 were affected by COVID during the first wave (mean age 63.0 ± 12.4 years, 27.3% female), while 60 during the second/third wave (mean age 65.7 ± 9.2 years, 25% female, *p* = 0.330 and *p* = 0.84 respectively vs first wave) and nobody was yet vaccinated. Of these, only the 7.6% of the participants did not required hospitalization during COVID disease. The 54.9% of patients were retired (42.9% and 63.3% in the first and second/third wave respectively, *p* = 0.05), 14% lives alone, and the 94.1% had a primary caregiver, intended as the person to refer in case of need (52.5% husband/wife/partner, 28.8% son/daughter, 7.9% other family members and 2.9% another person). No significant differences between socio-demographic variables besides except retirement (borderline significant) were observed between patients in the first and second/third waves.

In Table 1 are reported comorbidities/COVID-19 clinical management at T_0_ and clinical data/functional evaluation at T_1_ of the whole sample and according the two different waves. Comparing the first wave with the second/third one, there are some significant differences regarding comorbidities. Specifically, there are higher percentage of cardiopathy (*p* = 0.011), dyslipidemia (*p* = 0.024) and diabetes (*p* = 0.01) in the second/third waves. NIV treatment was significantly more present in the second/third wave group (*p* = 0.001). Conversely, BMI is higher in the first wave (27.82 ± 4.94 vs. 25.76 ± 4.69, *p* = 0.02). All significant results were confirmed controlling for the False detection rate at 5%, with borderline values for Dyslipidemia, BMI and Blood PH (adjusted *p* = 0.056).

At T_1_ evaluation, patients from the first wave showed a better clinical and functional profile as compared to patients from the second/third wave. Specifically, patients from the second/third wave had higher resting heart rate (*p* = 0.001), lower left ventricular ejection fraction (*p* = 0.002) and lower right ventricular function as assessed by TAPSE (*p* = 0.001). Moreover, bicarbonate was higher likely reflection a more compromised renal function. Although not statistically significant, patients from the second/third wave also showed a reduced distance covered at the 6MWT.

In Table 2 are reported GAD-7 and PHQ-9 percentage in the different waves and the comparison with the normative data. The total sample shows mild anxious (6.2 ± 5.45) and depressive (5.7 ± 5.4) symptoms. Comparing with normative data, both anxiety and depression symptoms are significantly higher than the normative sample [28,29].

At the NSESS, the 56.3% of the sample show mild to severe PTSD scores (32% mild, 14.6% moderate and 9.7% severe).

Among the correlations between psychological and clinical variables, a weak albeit significant association was observed only between GAD-7 and LVEF and between GAD-7 and Blood_PH (*r* = 0.22, *p* = 0.04 and *r* = −0.24, *p* = 0.04, respectively) and between NSESS and paCO2 (*r* = 0.26, *p* = 0.03), but these significances were not confirmed after controlling for the False detection rate at 5% (*p* > 0.34 all).

### 3.2. Qualitative Results

Concerning difficulties faced in returning to work, patients mainly complained about the increased sick leaves caused by COVID-19 disease and its sequelae, and the change or loss of their own job. In the second and third waves, the majority of participants affirmed to not be affected by this issue as they were in retirement (see Figure 2).

Regarding resources useful to cope with challenges posed by COVID-19 disease, participants of all waves mainly reported the support received by their own family, by the healthcare professional, as well as a positive and hopefully mindful approach. No differences were unveiled between the waves (Figure 3).

Concerning emotions felt by patients over the progression of pandemic situation, negative affects prevail during outbreaks in China and Italy, as well as during their own disease. During the present time (T_1_) the main affect is joy. Moreover, second/third waves patients experimented significantly more emotions connected with disgust thinking about the pandemic outbreak in China. Meanwhile, first wave patients reported significantly more emotions connected with fear during their own disease (Table 3a,b).

## 4. Discussion

The present multicentric study sheds light on the possible clinical, functional, psychological and emotional differences observed in patients affected by COVID-19 during the first and the second/third waves, from two-three months to one year after discharge.

Regarding the whole sample, the presence of comorbidities and mean BMI indicative of overweight (26.7 ± 4.9) are in line with the existing literature suggesting these characteristics as possible risk factors for hospitalization during COVID-19 disease [30]. The main observed clinical differences consisted in the presence of a higher number of comorbidities in the second/third wave compared to the first one, including, but not limited to cardiopathy, dyslipidemia and diabetes. Focusing on cardiopathy, the higher number detected in the second/third wave could be link also to the disrupting impact that COVID managing had on health care organization that affected the organization of care in the Hospitals and the Cardiology Divisions of many areas, specifically in Northern Italy [31]. Patients’ BMI was significantly higher in the second/third wave than in the first one. Furthermore, in the second/third wave the most frequently used treatment was noninvasive mechanical ventilation. These data show that patients in the second/third wave presented a more severe clinical profile, as already shown by [2]. On the contrary, no significant differences were found about functional (i.e., 6MWT) and psychological variables in the three waves. However, comparing psychological data (collected at T1) with those from the normative population, we found that our patients showed significantly higher scores in anxiety and depressive symptoms. Specifically, in our entire sample, depressive and anxiety symptoms were reported by the 21.2% and 25% of patients, respectively. Similarly, previous studies reported higher levels of these symptoms in COVID-19 patients than in normative population [13,32,33]. For instance, Mazza et al. (2020) reported depressive and anxiety symptoms in the 11.3% and in the 42.2% of the sample, respectively. Again, two other studies reported the presence of anxiety in the 22.2% [34] and in the 28.8% [35] of post COVID patients. Renaud-Charest et al., 2021 showed that 11%-28% of patients reported depressive symptoms three months after discharge. However, the results are not always unanimous: Vlake et al., (2021) reports a median HADS score of 4, not indicative of depressive symptoms, and they did not unveil any anxiety symptoms [36].

In addition, the 24.3% of the current sample reported moderate to severe PTSD symptoms. These findings are consistent with previous literature, with slightly differences on percentages reported. For instance, data by Mazza et al. (2020), reported the presence of PTSD in the 28.5% of the sample. Similarly, Cai et al. (2020) reported moderate to severe PTSD scores in around 31%, Ferraris et al. (2021) unveiled PTSD symptoms over clinical cut-off in the 34.4% of the sample. Wang et al. (2020) described PTSD symptoms only in the 8.1% of the sample.

Moreover, Brunoni et al. (2021) [37] did not found evidence of a pandemic-related worsening psychopathology in a Brazilian cohort thus, however they found a decrease along three wave-COVID of depression, anxiety and stress coherently with our data.

Overall, many of the differences concerning the above-mentioned results can be connected with different tools used in psychological assessment; thus, a consensus on most effective instruments is needed and suggested.

Concerning difficulties posed by COVID-19 disease in respect to work activity, in the first wave patients complained significantly more various changes in their own activities or the loss of work. This is in line with the organizational transformation triggered by the COVID-19 pandemic, in particular in the first period. For instance, a lot of activities turned to be full smart working and some people decided to completely change their own working life starting the so-called “great resignation” [38,39,40]. During the second/third wave, significantly more people stated to have nothing to say concerning work as they were not employed or already in retirement.

Focusing on protective factors, the three waves described the following resources without significant differences: families, perceived social support and positive mindset and attitudes. These ones were already known as factors able to help individuals to give meaning to life and to cope with adversities [41,42].

Moreover, the qualitative analysis of the patients’ feelings unveiled the presence of fear in all phases of pandemic, that is from the outbreak of pandemic in China to own disease. Considering that the Plutchik’s Wheel of Emotions Theory [25] describes fear as ranging from “apprehension” to “terror”, the presence of fear is coherent with the higher levels of anxious and depressive symptoms reported by the current sample compared with the normative population [28,29]. The emotion of fear is also significantly higher in patients of the first wave with respect to those experiencing the disease during the second and third waves. This finding can be explained by the fact that, from an evolutionary perspective, the “unknown” is often perceived as dangerous because it might potentially threaten own survival [41]. In this regard, specifically during the first wave, the clinical and social consequences of coronavirus can be seen without any doubt as an unknown threat, letting arising feelings of fear as an adaptive response of human beings [43]. On the contrary, during the follow-up, that can be considered the end of the most dangerous phase of the COVID-19 disease, the prevalent reported emotion was joy. This can be considered as a post-traumatic coping strategy to deal with the negative past experience, as suggested by positive psychology [44].

Another difference unveiled between waves is related to the emotion of “disgust”, significantly more reported by patients of the first wave and related to the origin of the pandemic in China. This finding might be explained from a sociological perspective: blaming an outgroup can be an effective coping strategy to face uncertainty and worries that were mainly experienced during the first wave. Some studies described similar scapegoating reactions as a strategy to regain a sense of control in the face of medical and socio-political uncertainty [45,46]. However, further studies are necessary to better understand this issue.

Overall considering this research, the main strengths include the comparison between the different pandemic waves which, to our knowledge, is scant in the previous literature, as well as the use of both a qualitative and quantitative approach that delves into the subjective emotional viewpoint of patients. The main limitations regard the cross-sectional nature of the study, which does not allow any interpretation of causality, thus, the increased levels of anxiety, depression and PTSD reported could have been partly caused also by the somatic diseases referred by patients, and not only by the circumstances related to COVID-19. Moreover, the limited size of the sample, and the characteristics of the population, such as old age and provenance did not allow the generalization of the results.

## 5. Conclusions

This study sheds light on some of the clinical differences characterizing the different pandemic waves, showing the presence of more comorbidities in the second/third wave compared to the first one. Moreover, our COVID-19 patients at follow up reported a distressed psychological state characterized by depressive, anxious and PTSD symptoms regardless the pandemic wave. This malaise seems to be subjectively counterbalanced by protective factors and inner resources (families, social support, positive mindset) which appear to be stable across the three pandemic waves. After recovery from COVID-19, joy flourished, suggesting a positive side given by the sense of survivor despite the negative experience.

Overall, these findings may enrich the knowledge on differences between COVID-19 pandemic waves and related patient’s subjective experience. Therefore, these contributions may provide useful suggestions to further research and to customize the taking care of the patient over time in the daily practice adopting a holistic approach which poses attention to resources and the emotional needs of each individual too. This appears of paramount importance in the current post-pandemic time where the healthcare systems are asked to manage the post and Long-COVID sequala in a tailored way in order to maximize outcome and reduce costs.

## Figures and Tables

**Figure 1 healthcare-11-02477-f001:**
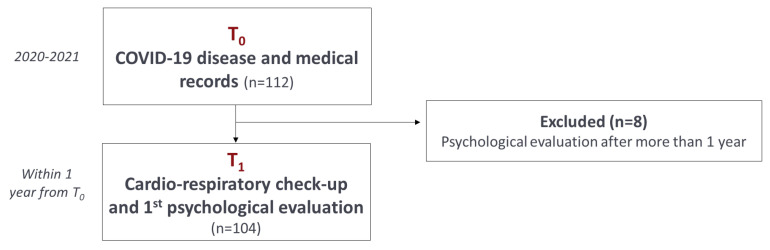
Flow diagram of case selection and research timing.

**Figure 2 healthcare-11-02477-f002:**
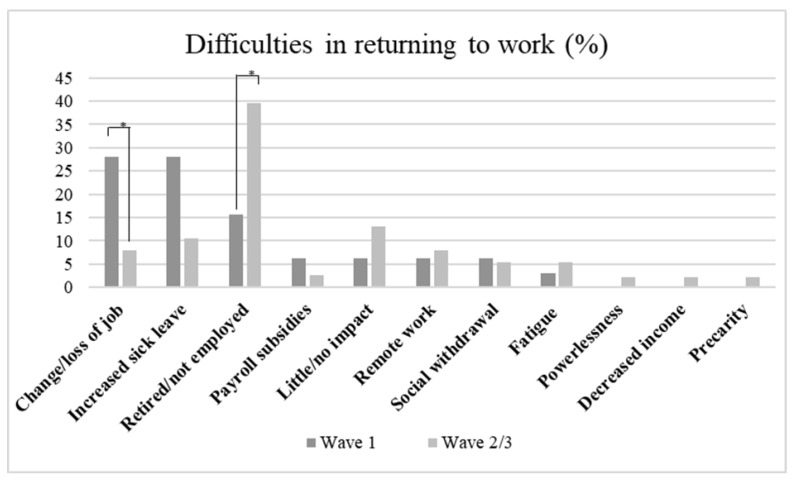
Percentage of difficulties faced in returning to work in different waves. * *p* < 0.05.

**Figure 3 healthcare-11-02477-f003:**
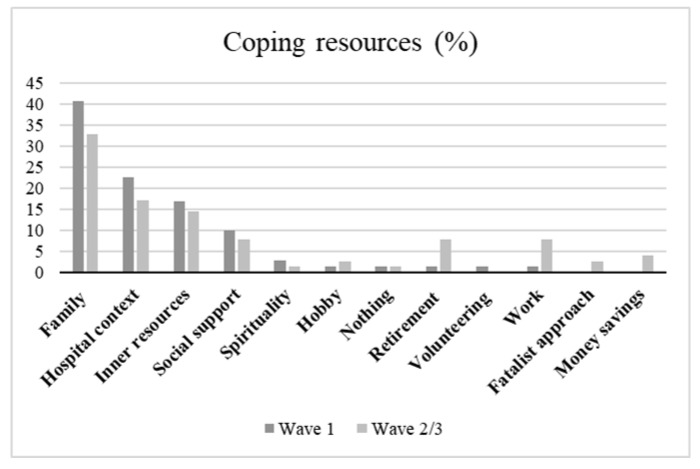
Percentage of coping resources reported in different waves.

**Table 1 healthcare-11-02477-t001:** Clinical variables (*n* = 104) in the first (*n* = 44) and second/third (*n* = 60) waves at T_0_ and T_1_.

T_0_	Total *n* (%)	1st Wave *n* (%)	2nd–3rd Wave *n* (%)	*p*
Comorbidities				
Hypertension	55 (52.9)	24 (54.5)	31 (51.7)	0.84
Cardiopathy	25 (24.0)	5 (11.4)	20 (33.3)	0.011 ‡
Dyslipidemia	26 (25.0)	6 (13.6)	20 (33.3)	0.024 †
Diabetes	19 (18.3)	3 (6.8)	16 (26.7)	0.011 ‡
COPD	7 (6.7)	3 (6.8)	4 (6.7)	1.00
Asma	8 (7.7)	4 (9.1)	4 (6.7)	0.72
OSA	9 (8.7)	4 (9.1)	5 (8.3)	1.00
Neoplasia	15 (14.4)	4 (9.1)	11 (18.3)	0.26
COVID-19 clinical management **				
OT	69 (66.3)	27 (61.4)	42 (70.0)	0.40
CPAP Therapy	51 (49.0)	20 (45.5)	31 (51.7)	0.56
NIV	20 (19.2)	1 (2.3)	19 (31.7)	<0.0001 ‡
IMV	18 (17.3)	6 (13.6)	12 (20.0)	0.44
**T** _1_	**Total** **(M ± DS)**	**1st wave** **(M ± DS)**	**2nd–3rd wave** **(M ± DS)**	** *p* **
Clinical data				
BMI	26.7 ± 4.9	27.8 ± 4.9	25.8 ± 4.7	0.02 †
Heart Rate	72.9 ± 11.8	66.3 ± 7.5	78.5 ± 12.0	<0.0001 ‡
LVEF	56.1 ± 10.6	59.0 ± 8.5	53.5 ± 11.6	0.002 ‡
TAPSE	22.9 ± 4.17	24.3 ± 3.18	18.1 ± 3.53	<0.0001 ‡
E/e’ ratio	9.4 ± 3.3	9.3 ± 3.6	9.4 ± 3.3	0.67
PAPs	31.7 ± 9.58	32.7 ± 7.1	31.3 ± 10.4	0.16
Blood PH	7.426 ± 0.030	7.415 ± 0.026	7.432 ± 0.031	0.024 †
PaO_2_ (mmHg)	76.7 ± 12.2	78.1 ± 9.6	75.9 ± 13.6	0.22
PaCO_2_ (mmHg)	37.4 ± 4.9	37.9 ± 4.0	37.1 ± 5.9	0.40
HCO_3_ (mmolL)	24.5 ± 2.1	23.7 ± 1.7	24.9 ± 2.3	0.013 ‡
Functional evalutation				
6MWT	396.4 ± 156.2	411.1 ± 156.1	383.1 ± 156.8	0.43

‡: significance (*p* < 0.05) confirmed controlling for the False detection rate at 5%; †: borderline significance (*p* = 0.056) controlling for the False detection rate at 5%; Abbreviations: COPD, Chronic Obstructive Pulmonary Disease; OSA, Obstructive Sleep Apnea; OT, Oxygen Therapy; CPAP, Continuous Positive Airway Pressure; NIV, Non Invasive Ventilation; IMV, invasive mechanical ventilation; BMI, Body Mass Index; LVEF, Left Ventricular Ejection Fraction; LVEDV, Left Ventricular End-Diastolic Volume; LVESV, Left Ventricular End-Systolic Volume; TAPSE, Tricuspid Annulus Plane Systolic Excursion; PAP, Pulmonary Artery Pressure; PaO_2_, Partial pressure of Oxygen; PaCO_2_, Partial pressure of carbon dioxide; HCO_3_, bicarbonates; 6MWT, 6 Minute Walking Test. ** patients can undergo to more than one treatment.

**Table 2 healthcare-11-02477-t002:** Psychological variables results, *n* (%), and comparison with normative data (χ^2^).

	Anxiety			Depression		
	No–Mild	Moderate–Severe		No–Mild	Moderate–Severe	
Normative data	4728 (94.0)	302 (6.0)		4682 (93.3)	336 (6.7)	
			(χ^2^) *p*			(χ^2^) *p*
Total sample	78 (75.0)	26 (25.0)	<0.0001	82 (78.8)	22 (21.2)	<0.0001
1st wave	28 (63.6)	16 (36.4)	<0.0001	31 (70.5)	13 (29.5)	<0.0001
2nd–3rd wave	50 (83.3)	10 (17.0)	0.003	51 (85.0)	9 (15.0)	0.002

**Table 3 healthcare-11-02477-t003:** (**a**) Patients’ emotions according to Plutchik’s Wheel of Emotions Theory. (**b**) Differences between waves concerning emotions experienced in the progression of pandemic.

(a)								
	What Were the Main EMOTIONS You Felt during:							
	Pandemic Outbreak in China		Pandemic Outbreak in Italy		Your Own Disease		The Present Time (Follow-Up)	
**1st wave**	Feeling no emotions = 11.54% 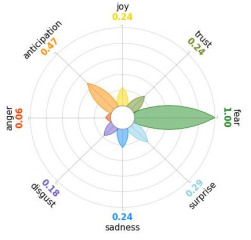		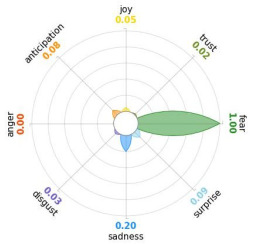		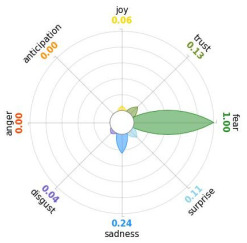		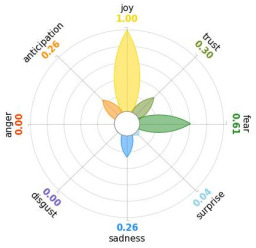	
**2nd/3rd waves**	Feeling no emotions = 10.45% 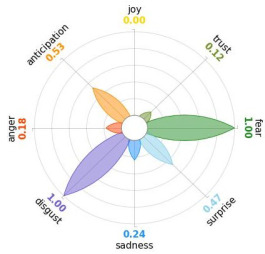		Feeling no emotions = 2.67% 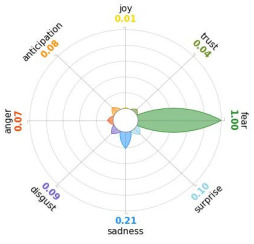		Feeling no emotions = 3.38% 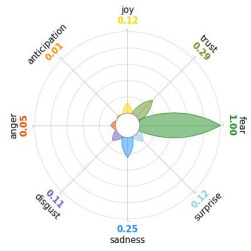		Feeling no emotions = 1.64% 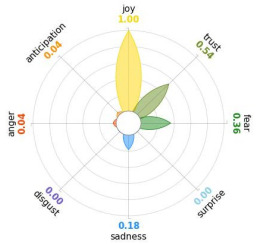	
(**b**)								
**Emotions**	**Pandemic Outbreak in China**	**Fisher Exact Test**	**Pandemic Outbreak in Italy**	**Fisher Exact Test**	**Your Own Disease**	**Fisher Exact Test**	**The Present Time (Follow-Up)**	**Fisher Exact Test**
Joy	40.35% vs. 0%	/	3.16% vs. 0.67%	0.302	40.35% vs. 6.08%	0.405	40.35% vs. 45.90%	0.581
Trust	7.69% vs. 2.99%	0.402	1.05% vs. 2.67%	0.651	8.04% vs. 14.19%	0.170	12.28% vs. 24.59%	0.102
Fear	32.69% vs. 25.37%	0.418	68.42% vs. 61.33%	0.277	63.39% vs. 49.32%	0.032 *	24.56% vs. 16.39%	0.361
Surprise	9.62% vs. 11.94%	0.773	6.32% vs. 6%	1	7.14% vs. 6.08%	0.803	1.75% vs. 0%	/
Sadness	7.69% vs. 5.97%	0.728	13.68% vs. 12.67%	0.847	15.18% vs. 12.16%	0.583	10.53% vs. 8.20%	0.757
Disgust	5.77% vs. 25.37%	0.006 **	2.11% vs. 5.34%	0.324	2.68% vs. 5.41%	0.36	0% vs. 0%	/
Anger	1.92% vs. 4.48%	0.631	0% vs. 4%	/	0% vs. 2.70%	/	0% vs. 1.64%	/
Anticipation	15.38% vs. 13.43%	0.797	5.26% vs. 4.67%	1	0% vs. 0.68%	/	10.53% vs. 1.64%	0.056

Note: * *p* < 0.05; ** *p* < 0.01.

## Data Availability

Inquiries can be directed to the corresponding author.

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
