# Peer review of "Patients’ Clinical and Psychological Status in Different COVID-19 Waves in Italy: A Quanti-Qualitative Study"

_healthcare, 2023, doi:10.3390/healthcare11182477_

Round 1

Reviewer 1 Report

I recommend the manuscript entitled "Patients’ Clinical and Psychological Status in Different COVID-19 Waves: A Quanti-Qualitative Study" in the International Journal of Environmental Research and Public Health. The study reported in the manuscript concerns the topical problem of the physical and mental well-being of patients during the waves of the COVID-19 epidemic. The authors describe the emotions and increased symptoms of anxiety, depression and PTSD experienced by patients. I would like to point out that the authors surveyed patients with COVID-19 who had somatic comorbidities that may have significantly affected their psychological status. I believe that the authors should mention in the Conclusions and the Limitations sections that the increased levels of anxiety, depression and PTSD observed in the patients could have been partly caused by the somatic diseases they reported, and not solely by the circumstances related to COVID-19.

Reviewer 3 Report

The authors investigate the important Psycological status aspect of COVID-19. Although interesting, I believe it is unthinkable to asses such an argument on such a small sample size. Moreover, vaccination status is completely ignored and this surely has had effects on clinical status as well as psychological.   

Reviewer 4 Report

The present study is aimed at comparing the psychophysical symptoms of COVID-19 patients infected in different waves and undergoing a functional, cardiological and pneumological check-up from 2-3 months to 1 year after recovery, adopting a mixed-method approach.

The topic is potentially interesting and I appreciate the qualitative method used to characterize the emotion felt by patients over the progression of the pandemic.

However the manuscript requires major changes to be suitable for publication. 

INTRODUCTION:

1.     It is not clear if the patients enrolled were all previously hospitalized patients. Since COVID-19 infection lead to very variable clinical manifestations it is crucial to clearly specify study setting, both in the introduction and in the method sections. In this case, it is also useful to coherently focus the background description on the post-acute physical and psychological sequelae of COVID-19 disease in hospitalized patients.

2.     Please better articulate the knowledge gap cover by the present study and the aims. Specifically, at the end of the section the author mentioned the topic of the relationship between COVID-19 physical symptoms and emotional consequences (lines 70-72 page 2), however in the aim and methods it is not clear how this issue is addressed.

MATERIAL AND METHODS:      

3.     I suggest to modify the organization of the paragraphs to improve readers’ understanding of the methodology. It could be useful to include a “Participants and procedure” paragraph including all the relevant information to understand the overall study design. To this end, please move here the description of the subgroup considered in the present study (patients infected during the first wave VS those infected during second and third waves), which in the present version of the manuscript is reported in the statistical analysis paragraph. Moreover clarify here if all of the participants were hospitalized at T0 due to COVID-19 (see comment 1).

4.     In the exclusion criteria, how do you mean for “acute severe clinical conditions”? It could be useful to add some examples.

5.     I suggest to add a flow chart of case selection, as required when reporting results from observational study.

6.     In the Materials paragraph the instrument administered at T1 are well-described, while the description of T0 measurements retrospectively collected from clinical records is completely missing and only inferred by Table 1 in the Results.

7.     In the statistical analysis paragraph (page 4, lines 156-157), it is mentioned an analysis of “association between couple of variables”. Which variables? Why?

8.     Please specify if themes categorization for text analysis (page 4, lines 170-173) was conducted independently by the 2 researchers and how they deal with discrepancies.

RESULTS

9.     Please specify any eventual significant differences on socio-demographic variables between groups.

10.  The authors stated that 94.1% of the sample had a primary caregiver. The use of this term in this context is quite surprising, since the average age of the sample is of 65.7 years. Could you please briefly explain what do you mean with this term? If I read this term I guess that almost the entire sample require high assistance in daily activities.

11.  It could be useful to report also in Table 1 which significance survived to the FDR correction.

12.  Some correlation are reported between variables (page 6, lines 240-243), but without neither a previous explanation of the rationale and variables used (see comment 7) nor an interpretation of findings in the Discussion.

13.  The definition of Table 3 should be improved.

14.  Overall comment: I can see that there are important intrinsic differences in the groups physical status at baseline. I guess that also some important differences for socio-demographic variables are present, since the majority of the participants in the second/third waves group are retired (VS first wave ones, page 6 lines 245-249). However, the author interpret all the findings only in light of exposure to COVID-19 in different period, although they didn’t control for these baseline differences in the analysis performed.

DISCUSSION: The discussion describe (some) findings in light of the available literature, without clear and focused answers to study questions. The practical implication of the findings and the recommendation for future research are not reported.

Round 2

Reviewer 3 Report

The authors investigate the important Psycological status aspect of COVID-19. Although interesting, as already mentioned in the previous revision, I believe it is unthinkable to asses such an argument on such a small sample size. Moreover, vaccination status is completely ignored and this surely has had effects on clinical status as well as psychological.   
